# Trends, geographical variation and factors associated with prescribing of gluten-free foods in English primary care: a cross-sectional study

Alex J Walker, Helen J Curtis, Seb Bacon, Richard Croker, Ben Goldacre

EBM DataLab, Centre for Evidence Based Medicine, Nuffield Department of Primary Care Health Sciences, University of Oxford, Oxford, UK

**Correspondence to**
Dr Ben Goldacre;
ben.goldacre@phc.ox.ac.uk

## ABSTRACT

**Objectives** There is substantial disagreement about whether gluten-free foods should be prescribed on the National Health Service. We aim to describe time trends, variation and factors associated with prescribing gluten-free foods in England.

**Setting** English primary care.

**Participants** English general practices.

**Primary and secondary outcome measures** We described long-term national trends in gluten-free prescribing, and practice and Clinical Commissioning Group (CCG) level monthly variation in the rate of gluten-free prescribing (per 1000 patients) over time. We used a mixed-effect Poisson regression model to determine factors associated with gluten-free prescribing rate.

**Results** There were 1.3 million gluten-free prescriptions between July 2016 and June 2017, down from 1.8 million in 2012/2013, with a corresponding cost reduction from £25.4 million to £18.7 million. There was substantial variation in prescribing rates among practices (range 0 to 148 prescriptions per 1000 patients, IQR 7.3–31.8), driven in part by substantial variation at the CCG level, likely due to differences in prescribing policy. Practices in the most deprived quintile of deprivation score had a lower prescribing rate than those in the highest quintile (incidence rate ratio 0.89, 95% CI 0.87 to 0.91). This is potentially a reflection of the lower rate of diagnosed coeliac disease in more deprived populations.

**Conclusion** Gluten-free prescribing is in a state of flux, with substantial clinically unwarranted variation between practices and CCGs.

## INTRODUCTION

The prescribing of gluten-free foods is currently a controversial issue. Adherence to a gluten-free diet is the only effective treatment for coeliac disease and such prescribing may be associated with better adherence,[1] but there is currently extensive discussion about whether it is cost-effective as an intervention,[2] given that gluten-free foods are increasingly readily available and affordable.

Currently, National Health Service (NHS) patients in England with a confirmed diagnosis of gluten-sensitive enteropathy can

receive a wide range of gluten-free foods on an NHS prescription. These are subject to normal prescription charges and exemptions. Patients who do not have a confirmed diagnosis are not eligible for treatment. NHS England have recently consulted on whether to restrict gluten-free prescribing and states that they are 'supportive of restricting the availability of gluten-free foods on prescription'.[3] The National Institute for Health and Care Excellence (NICE) guideline development group in March 2016 recommended that "people with coeliac disease and their family/carers should be aware of, and have access to, gluten-free food prescriptions to support adherence to a gluten-free diet",[4] and national prescribing guidelines recommend the prescription of staple foods, such as bread and flour.[5] Clinical Commissioning Groups (CCGs), which are responsible for commissioning of healthcare services in their local area, currently have diverse prescribing policies for gluten-free foods.[6] More recently, the NICE Quality Standard, stated "healthcare professionals should highlight if gluten-free food products are available on prescription to help people to maintain a gluten free diet",[7] which highlights the risk of inequality

## Strengths and limitations of this study

► We were able to measure the prescribing of gluten-free foods across all prescribing in England, eliminating bias. We also removed seasonal variation by aggregating savings over 12 months.

► As well as gluten-free prescribing variation at practice and Clinical Commissioning Group level, we have described long-term prescribing trends at national level, back to 1998.

► Using the available data, we were unable to look at gluten-free prescribing at prescriber level or investigate factors associated with prescribing to individual patients.

of access to gluten-free foods. In contrast to in England, recommendations for prescribing of gluten-free foods in Northern Ireland, Scotland and Wales are broadly in line with national prescribing guidelines, and in Scotland, gluten-free food prescribing is managed by the community pharmacy–based Scottish Gluten-free Food Service.[6]

We set out to describe trends over time and national variation in gluten-free prescribing, and assess factors associated with the prescribing rate of gluten-free foods in English general practices. This will allow for accurate assessment and forecasting of costs, and better understanding of the range of current norms in clinical practice, the reasons for variation in prescribing choices and the factors that may help to modify future prescribing behaviour.

## METHODS
### Study design
This is a retrospective cohort study incorporating English general practitioner (GP) practices, measuring variation in prescribing of gluten-free foods over time, geographically and determining what factors are associated with volume of gluten-free prescribing. We use both mixed-effects Poisson regression and mixed-effects logistic regression to investigate correlation of gluten-free prescribing with various practice characteristics.

### Setting and data
We used annual prescription cost analysis (PCA) data in order to describe long-term trends in gluten-free prescribing (between 1998 and 2016). These datasets were downloaded from NHS Digital or National Archives (https://digital.nhs.uk/article/4214/Prescribing) and are at individual drug level, but aggregated at a national level rather than practice level.

We also used data from our OpenPrescribing.net project, which imports prescribing data from the monthly data files published by NHS Digital.[8] These contain data on cost and volume prescribed for each drug, dose and preparation, for each month, for each English general practice. We extracted the most recent year of data available (year to June 2017 inclusive). This allowed us to define gluten-free prescribing and generate the composite prescribing measure described below. We also matched the prescribing data with publicly available data on practices from Public Health England.[9] This allowed us to stratify the analysis to look at reasons for variation in gluten-free prescribing at the practice level. All standard English practices labelled within the data as a 'GP practice' were included within the analysis; this excluded prescribing in non-standard settings such as prisons. Additionally, in order to further exclude practices that are no longer active, those without a 2015/2016 Quality Outcomes Framework (QOF) score and those with a list size (ie, number of patients registered at the practice) under 1000 were excluded. Using inclusive criteria such as this reduced the likelihood of obtaining a biased sample.

### Long-term prescribing trends
We used the PCA data to describe the long-term trends in gluten-free prescribing items per 1000 patients (see online supplementary appendix A for code list) at national level for each year between 1998 and 2016. These were aggregated by type of gluten-free product into the categories: bread, biscuits, mixes/grains/flours, pasta, cereals and other. The total number of gluten-free prescriptions was determined, which was then converted to a rate per 1000 people, using the mid-year national population estimates for England.[10] We created a stacked line graph to represent these data.

### Geographical variation at CCG level
We determined the rate of gluten-free prescribing, defined as the number of gluten-free items prescribed (see online supplementary appendix A for code list) divided by the mean practice list size over the most recent year. These rates were aggregated by grouping each practice to its parent CCG and described using summary statistics, a histogram and a map in which each CCG's prescribing was represented using a colour spectrum.

### Monthly trends and variation across practices
We generated descriptive statistics to describe the rate of gluten-free prescribing per month within the cohort, including medians and IQRs. We then described the monthly trends between July 2012 and June 2017 by calculating deciles at the practice level for each month and plotting these deciles. We also used a histogram to describe the distribution of gluten-free prescribing volume among practices.

### Factors associated with gluten-free prescribing
We used the rate of gluten-free prescribing per 1000 patients, aggregated over the previous year as the outcome variable, and defined a number of other explanatory variables for the purpose of determining which factors are associated with gluten-free prescribing, as follows.

We have previously developed a composite measure score to describe overall prescribing quality. The 36 current standard prescribing measures on OpenPrescribing.net[11] have been developed to address issues of cost, safety or efficacy by doctors and pharmacists working in collaboration with data analysts. Each month, OpenPrescribing calculates the centile that each practice is in, for each measure. Measures are oriented such that a higher percentile corresponds to what would be considered 'worse' prescribing (with the exception of those where no value judgement is made, ie, direct-acting oral anticoagulants[12] and pregabalin,[13] which are excluded from this analysis). For the purpose of this study, we calculated the mean percentile that each practice was in across all measures, over the latest available 6 months of data, as a composite measure of prescribing quality.

We also used a number of practice demographic metrics (available from Public Health England)[9]: QOF score, which is a performance management metric used for

GPs within the NHS, produced by NHS Digital; practice list size (calculated as mean over the most recent year); Index of Multiple Deprivation (IMD) score; patients with a long-term health condition (%); patients over 65 (%) and whether each practice is a 'dispensing practice' with an in-house pharmacy service (yes or no).

We stratified the rate of gluten-free prescribing according to the factors defined above. These factors were also entered into a Poisson regression model, then a mixed-effects Poisson regression model with gluten-free prescribing rate as the dependent variable, the above variables as fixed-effect independent variables and the CCG of each practice as a random-effect variable. Prescribing and other practice quality measures were divided a priori into quintiles for analysis, except for existing binary variables (ie, dispensing practice). Practices with missing data for a particular variable were not included in models containing that variable. From the resulting model, incidence rate ratios were calculated, with corresponding 95% CIs. The level of missing data was determined and reported for each variable.

### Software and reproducibility
Data management was performed using Python, with analysis carried out using Stata V.13.1. Data and all code can be found online (https://figshare.com/s/79d6ada8b8ecb4cfdce9).

## RESULTS
### Summary of population characteristics
There were 8185 practices within the prescribing dataset labelled as a 'GP practice'. A total of 558 practices were excluded as they did not have a QOF score or had a list size under 1000. Therefore, 7627 standard current practices were included within the study (table 1).

**Table 1** Practice summary characteristics

| | Value | IQR | |
|---|---|---|---|
| Total practices | 7627 | – | – |
| Median gluten-free items prescribed within previous year | 115 | 40 | 240 |
| Median rate of gluten-free prescribing (per 1000 patients) | 17.8 | 7.4 | 31.9 |
| Median composite measure score (lower is better) | 46.1% | 40.5% | 51.5% |
| Median QOF score (maximum possible score 559) | 546 | 529.0 | 555.2 |
| Median practice list size | 6875 | 4244 | 10254 |
| % of dispensing practices | 13.4 | – | – |
| Median % of patients with long-term health conditions | 17.2 | 12.1 | 21.4 |
| Median % of patients over 65 years | 53.4 | 48.3 | 58.5 |

QOF, Quality Outcomes Framework.

There were 1.3 million gluten-free prescriptions nationally over the year between July 2016 and June 2017, with a total expenditure of £18.7 million (mean—£14.41 per prescription item). This has decreased from 1.8 million prescriptions and £25.4 million during the year July 2012 to June 2013. If the spending on gluten-free prescribing were to continue at the level in the most recent month (June 2017), this would result in an annual expenditure of £15.6 million. The level of missing data was low, with 99.5% of practices having complete data for all variables (see online supplementary appendix B).

### Long-term national trends in gluten-free prescribing
From the PCA data, a broad increase in gluten-free dispensing was observed between 1998 and 2010, followed by a decline in most categories between 2010 and 2016 (figure 1). However, there was an increase in cereal products dispensed between 2011 and 2015, following their addition to guidance by the Advisory Committee on Borderline Substances.

### Variation between CCGs
Among CCGs, variation in gluten-free prescribing rate varied from 0.1 to 55.5 per 1000 patients over the last year, with a median value of 20.0 (IQR 12.5–30.9). The distribution over the most recent year is represented by a histogram in figure 2 and as a map in figure 3 (with an interactive version available at the URL here).

### Monthly trends and variation across practices
We further investigated the decline in prescribing observed at the end of the national data (figure 1) by measuring variation across practices using the more detailed monthly datasets (figure 4). Here, the median prescriptions per month remained steady at around 2 per 1000 patients per month between July 2012 and mid-2015, then declined to 1.04 per 1000 patients in June 2017. The 10th centile of practices remained at zero throughout this period, with over 20% of practices having zero prescriptions from February 2017 onwards.

Aggregated prescribing figures over the most recent year of data show that almost all practices (92.7%) had at least one gluten-free prescription within the last year (figure 5). The median number of prescription items was 115 (IQR 40–240), with a median rate of 17.8 per 1000 patients over the year (IQR 7.3–31.8). The maximum rate of prescribing for any practice was 148.1 per 1000 patients per year.

### Factors associated with gluten-free prescribing rate
Using the mixed-effect Poisson regression model, we found the rate of gluten-free prescribing to be significantly associated with each of the factors investigated, with the exception of the percentage of patients with a long-term health condition (table 2). Practices with a lower (better) score in our composite prescribing measure had a lower rate of gluten-free prescribing than practices in higher categories. For both QOF score and practice list size, the top four quintiles appeared to

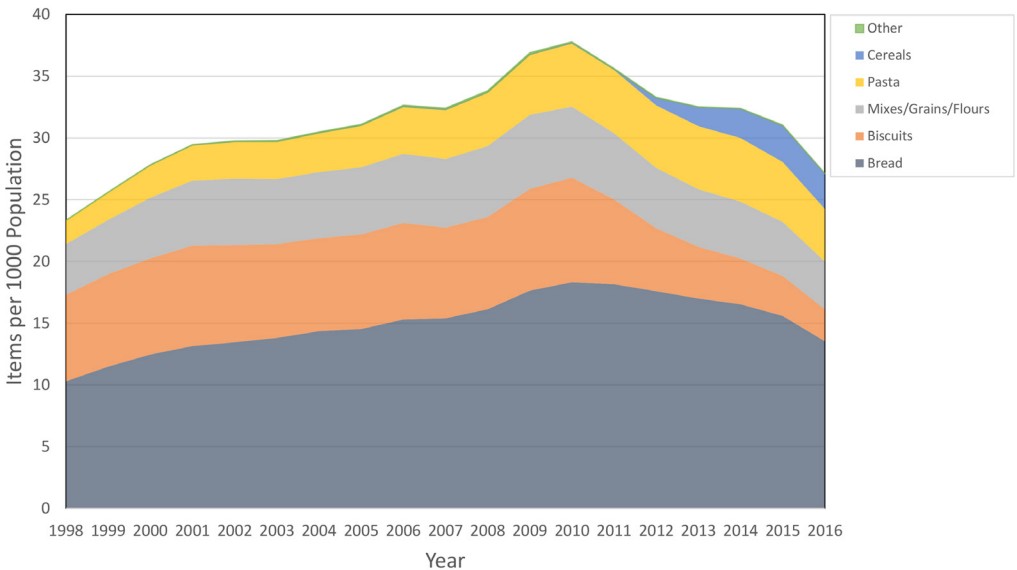

**Figure 1** Stacked line graph to show gluten-free food items dispensed in England per 1000 population, grouped into categories. 'Other' includes cakes/pastries, meals and cooking aids.

have a similar rate of prescribing, but the lowest quintile (ie, practices with the lowest QOF score or smallest list size) was significantly lower in prescribing rate than the higher (better performing) quintiles. IMD score had a dose-response relationship with gluten-free prescribing rate, with the most deprived quintile of practices prescribing at a 11% lower rate than those in the least deprived areas (incidence rate ratio (IRR) 0.89, 95% CI 0.87 to 0.91). Dispensing practices had a higher rate of prescribing than non-dispensing practices (IRR 1.7, 95% CI 1.06 to 1.09). The strongest effect sizes were observed with age, where practices with the greatest proportion of patients over 65 prescribed at a 46% higher rate than those with the lowest proportion (IRR 1.46, 95% CI 1.42 to 1.50). We found that CCG (as a random effect) was significantly associated with rate of

gluten-free prescribing (likelihood-ratio test vs Poisson regression, P<0.0001).

## DISCUSSION
### Summary of findings
We determined that overall prescribing of gluten-free foods rose from 1998 to 2010, but has declined substantially since this peak. We found a high level of variation in rate of prescribing between practices, some of which is driven at the CCG level, where there is also a great deal of variation. This variation has remained broadly similar over time. It was also possible to further explore the reasons for this variation through modelling. Here, we found that practices in the most deprived areas had a significantly lower rate of gluten-free prescribing than those in less deprived areas, and that practices performing poorly on a composite measure of prescribing quality were more likely to prescribe gluten-free foods. We also found age distribution and practice size to be important determinants of variation in gluten-free prescribing rate, among other factors.

### Strengths and weaknesses
We included all typical practices in England, thus minimising the potential for obtaining a biased sample. We were also able to use real prescribing and spending data, which are sourced directly from pharmacy claims and therefore did not need to rely on the use of surrogate measures. We also used prospectively gathered prescribing data rather than survey data, eliminating the possibility of recall bias. We excluded a small number of practices without a QOF score, as many of these practices are no longer active and we reasoned that any practice not participating in QOF would be less representative of a 'typical' GP practice. This may have excluded a small number of practices that opened since the 2015/2016 QOF scores were calculated; however,

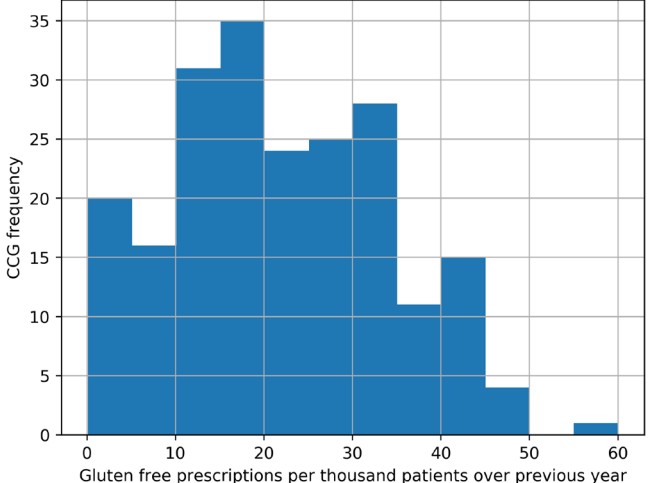

**Figure 2** Distribution among Clinical Commissioning Groups (CCGs) of the number of gluten-free prescriptions per 1000 patients between July 2016 and June 2017.

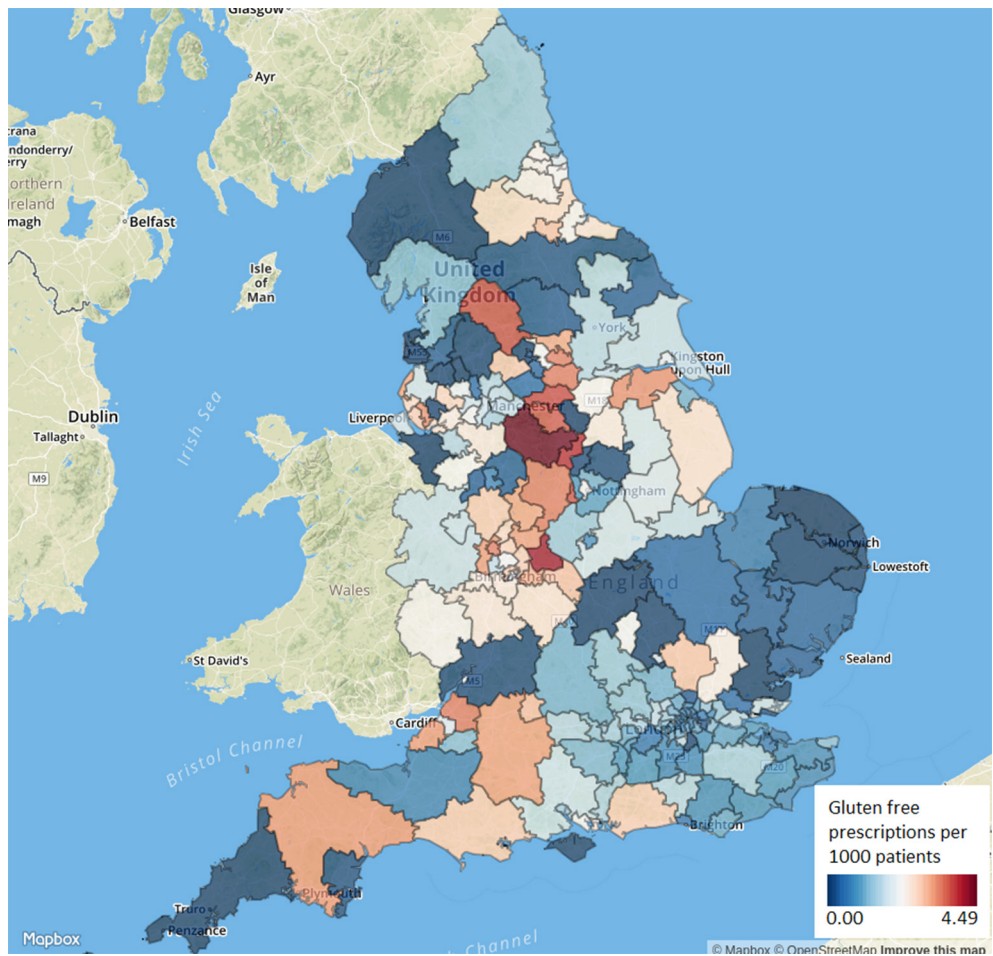

**Figure 3** Variation in gluten-free prescribing between Clinical Commissioning Groups in England, number of prescriptions per 1000 patients, for June 2017.

there are no grounds to believe that such practices would have been systematically different to the rest of our population with respect to gluten-free prescribing or factors associated with it. Due to a large sample size and large effect sizes, we obtained a high level of statistical significance in many of the associations we observed.

### Findings in context

Among the other possible reasons for variation in gluten-free prescribing that were not able to be measured here, an important factor is the incidence of coeliac disease. West *et*

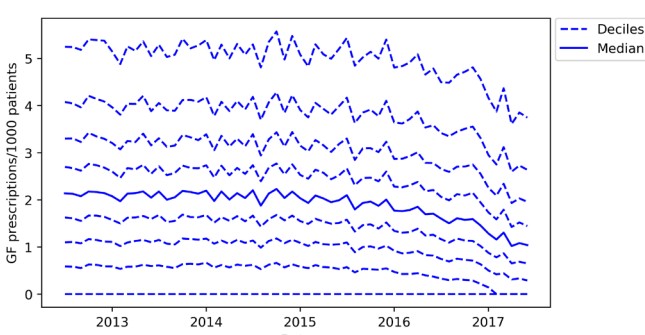

**Figure 4** Practice deciles of gluten-free (GF) prescriptions per thousand patients, for each month from July 2012 to June 2017.

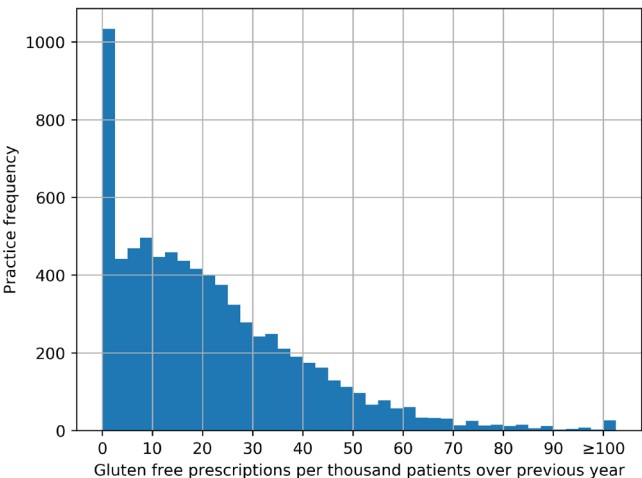

**Figure 5** Distribution among practices of the total number of gluten-free prescriptions per 1000 patients between July 2016 and June 2017 (values over 100 are aggregated into the final column).

**Table 2** Absolute rates of gluten-free prescribing per 1000 patients, stratified by various factors, followed by incidence rate ratios from a univariable Poisson model, then from a multivariable mixed-effects Poisson model

| | Quintile boundaries | Median gluten-free prescriptions per 1000 patients | Univariable incidence rate ratio* | 95% CI | Multivariable incidence rate ratio† | 95% CI |
|---|---|---|---|---|---|---|
| Composite measure score (lower percentile is better) | <39% | 13.44 | Reference | | Reference | |
| | 39% to 44% | 16.18 | 1.08 | 1.06 to 1.10 | 1.08 | 1.06 to 1.09 |
| | 44% to 48% | 18.39 | 1.15 | 1.13 to 1.17 | 1.09 | 1.07 to 1.10 |
| | 48% to 53% | 19.29 | 1.18 | 1.16 to 1.20 | 1.09 | 1.07 to 1.11 |
| | >53% | 21.68 | 1.27 | 1.25 to 1.29 | 1.13 | 1.10 to 1.15 |
| Quality Outcomes Framework score (maximum=559) | <522 | 14.13 | Reference | | Reference | |
| | 522 to 541 | 16.92 | 1.17 | 1.15 to 1.19 | 1.10 | 1.08 to 1.12 |
| | 541 to 550 | 18.13 | 1.20 | 1.18 to 1.22 | 1.10 | 1.08 to 1.12 |
| | 550 to 556 | 19.45 | 1.26 | 1.24 to 1.28 | 1.09 | 1.07 to 1.11 |
| | >556 | 20.35 | 1.29 | 1.27 to 1.31 | 1.11 | 1.09 to 1.13 |
| Index of Multiple Deprivation | Least deprived | 19.46 | Reference | | Reference | |
| | – | 19.67 | 1.00 | 0.98 to 1.01 | 0.96 | 0.94 to 0.98 |
| | – | 17.36 | 0.93 | 0.91 to 0.94 | 0.95 | 0.93 to 0.97 |
| | – | 14.93 | 0.82 | 0.81 to 0.83 | 0.87 | 0.85 to 0.89 |
| | Most deprived | 17.62 | 0.94 | 0.92 to 0.95 | 0.89 | 0.87 to 0.91 |
| Practice list size | <3784 | 12.35 | Reference | | Reference | |
| | 3785 to 5796 | 17.49 | 1.16 | 1.14 to 1.17 | 1.18 | 1.16 to 1.20 |
| | 5798 to 8018 | 18.50 | 1.15 | 1.13 to 1.17 | 1.18 | 1.16 to 1.20 |
| | 8020 to 11165 | 20.21 | 1.23 | 1.21 to 1.25 | 1.19 | 1.17 to 1.21 |
| | >11165 | 19.11 | 1.16 | 1.14 to 1.18 | 1.17 | 1.15 to 1.19 |
| Dispensing practice? | No | 17.57 | Reference | | Reference | |
| | Yes | 19.44 | 1.06 | 1.05 to 1.08 | 1.07 | 1.06 to 1.09 |
| % of patients over 65 | <10.8% | 10.29 | Reference | | Reference | |
| | 10.8% to 15.4% | 16.29 | 1.28 | 1.25 to 1.30 | 1.09 | 1.07 to 1.11 |
| | 15.4% to 18.8% | 19.31 | 1.47 | 1.44 to 1.49 | 1.22 | 1.20 to 1.25 |
| | 18.8% to 22.4% | 22.39 | 1.65 | 1.63 to 1.68 | 1.34 | 1.31 to 1.37 |
| | >22.4% | 22.83 | 1.68 | 1.65 to 1.70 | 1.46 | 1.42 to 1.50 |
| % of patients with a long-term health condition | <47.0% | 12.82 | Reference | | Reference | |
| | 47.0% to 51.5% | 17.34 | 1.19 | 1.17 to 1.20 | 0.99 | 0.97 to 1.00 |
| | 51.5% to 55.3% | 17.87 | 1.26 | 1.24 to 1.28 | 1.00 | 0.98 to 1.01 |
| | 55.4% to 59.7% | 21.26 | 1.36 | 1.34 to 1.38 | 0.99 | 0.97 to 1.01 |
| | >59.7% | 21.04 | 1.34 | 1.32 to 1.36 | 0.98 | 0.96 to 1.00 |

*Poisson regression.
†Mixed-effects Poisson regression, adjusted for all other variables in table as fixed effects, plus Clinical Commissioning Group as a random effect.

al[14] describe the national variation in coeliac disease prevalence at a regional level, but this is too broad a level to meaningfully adjust for in our analysis. Additionally, it seems likely that the reasons for variation in coeliac disease prevalence would be somewhat similar to those for variation in gluten-free prescribing. For example, IMD is known to be associated with coeliac disease incidence and prevalence.[14–16] We also found that percentage of patients over 65 is strongly associated with gluten-free prescribing, which is unsurprising given that coeliac disease prevalence increases with age.[14]

We found CCGs to be a significant driver of variation, with a large variation in gluten-free prescribing at the CCG level, and a significant effect of CCG identifier within our mixed-effect modelling. This is likely due to variations in CCG policies and therefore strongly suggests that practices are responsive to CCG prescribing guidance, at least on the issue of gluten-free food. CCG policies range from following national guidelines (which recommend prescribing of staple foods)[5] to a partial or complete withdrawal of prescriptions.

### Policy implications and interpretation
Gluten-free prescribing is clearly in a state of flux at the moment, with an apparent rapid reduction in prescribing

nationally. This may be viewed as a positive change, for example by NHS England,[17] freeing up resources to be more effectively used elsewhere. However, groups that represent patients with coeliac disease, such as Coeliac UK,[18] are strong advocates for the continued prescribing of gluten-free foods. Although gluten-free foods are perceived to be becoming cheaper and more widely available, they remain more expensive than budget wheat-containing options (on average five times greater),[19] and it is argued that vulnerable populations may struggle to source appropriate foods for their condition without prescriptions. Any potential future reductions in prescribing would therefore be controversial. However, it is clear that the level of variation in gluten-free prescribing is very high, and that this variation appears to exist largely without good reason, being determined to a large extent by factors such as CCG.

## Summary

Prescribing of gluten-free foods is declining rapidly and given recent policy changes seems likely to continue to do so. The variation in prescribing between different practices and in different areas of the country seems to be largely determined by the decisions and preferences of clinicians and local health services.

**Contributors** AJW and BG conceived and designed the study. AJW and HJC collected and analysed the data with methodological and interpretation input from RC, SB and BG. AJW drafted the manuscript. All authors contributed to and approved the final manuscript. SB was lead engineer on the associated website resource with input from RC, AJW, BG and HJC. BG supervised the project and is guarantor. Lead engineer on the original OpenPrescribing tool was Anna Powell-Smith.

**Funding** This work was supported by the NIHR Biomedical Research Centre, Oxford; the Health Foundation grant (Unique Award Reference Number 7599); and by a National Institute for Health Research (NIHR) School of Primary Care Research (SPCR) grant (ref no. 327). No specific funding was sought for the analysis reported in this paper.

**Disclaimer** Funders had no role in the study design, collection, analysis and interpretation of data; in the writing of the report; and in the decision to submit the article for publication.

**Competing interests** BG has received research funding from the Laura and John Arnold Foundation, the Wellcome Trust, the Oxford Biomedical Research Centre, the NHS National Institute for Health Research School of Primary Care Research, the Health Foundation and WHO; he also receives personal income from speaking and writing for lay audiences on the misuse of science. AJW, HJC, SB and RC are employed on BG's grants for the OpenPrescribing project. RC is employed by a CCG to optimise prescribing and has received (over 3 years ago) income as a paid member of advisory boards for Martindale Pharma, Menarini Farmaceutica Internazionale SRL and Stirling Anglian Pharmaceuticals Ltd.

**Patient consent** Not required.

**Ethics approval** This study uses exclusively open, publicly available data; therefore, no ethical approval was required.

**Provenance and peer review** Not commissioned; externally peer reviewed.

**Data sharing statement** All analytic data and code are available online (https://figshare.com/s/39a4301a29316bc86b35).

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
