## [Reviewer comments · BMJ Open]

ARTICLE DETAILS

TITLE (PROVISIONAL)	Trends, geographic variation, and factors associated with prescribing of gluten-free foods in English primary care: a cross sectional study
AUTHORS	Walker, Alex; Curtis, Helen; Bacon, Seb; Croker, Richard; Goldacre, Ben

VERSION 1 – REVIEW

REVIEWER	Federico Biagi University of Pavia, Italy
REVIEW RETURNED	08-Jan-2018

GENERAL COMMENTS	In this paper Walker et al. studied trends, geographic variation, and factors associated with prescribing of gluten-free foods in England. I have got the following comments: 1. Non-English readers will find very difficult to understand and to appreciate this paper. Since BMJ is an international journal, a brief introductory description of how gluten-free products are prescribed in the UK will help. For example, are the GF products totally free for the patients once prescribed or does the patients have to pay a part of it? Can the patients choose what they are prescribed? Can gluten-free products be prescribed only to coeliac patients? What about patients “said” to be affected by gluten sensitivity?2. Did the authors try to evaluate the role of tertiary referral centres? In other words, I would expect that the prescription of gluten free-products is higher near Sheffield, Derby, etc were very important tertiary referral centres are based.
---

REVIEWER	Dr Matthew Kurien Department of Infection, Immunity & Cardiovascular Disease, University of Sheffield, UK
REVIEW RETURNED	23-Jan-2018

GENERAL COMMENTS	I thoroughly enjoyed reading this well written manuscript and commend the authors for undertaking this interesting and important work. I have 2 minor comments that the authors may wish to consider: 1) I think it would be useful within the Introduction to emphasise this differences in prescribing practices between England and the devolved nations.2) Please could the authors provide complete details for reference 1.
--

VERSION 1 – AUTHOR RESPONSE

Reviewer: 1

In this paper Walker et al. studied trends, geographic variation, and factors associated with prescribing of gluten-free foods in England. I have got the following comments:

1. Non-English readers will find very difficult to understand and to appreciate this paper. Since BMJ is an international journal, a brief introductory description of how gluten-free products are prescribed in the UK will help. For example, are the GF products totally free for the patients once prescribed or does the patients have to pay a part of it? Can the patients choose what they are prescribed? Can gluten-free products be prescribed only to coeliac patients? What about patients "said" to be affected by gluten sensitivity?

- Thanks for this useful suggestion, we have added a section in the introductory paragraph to describe this: "Currently, NHS patients in England with a confirmed diagnosis of gluten-sensitive enteropathy can receive a wide range of gluten-free foods on an NHS prescription. These are subject to normal prescription charges and exemptions. Patients who do not have a confirmed diagnosis are not eligible for treatment."

2. Did the authors try to evaluate the role of tertiary referral centres? In other words, I would expect that the prescription of gluten free-products is higher near Sheffield, Derby, etc were very important tertiary referral centres are based.

- This is an interesting observation, while in theory primary care prescribing should not be directly affected by the presence of nearby tertiary referral centres, it is possible that such centres could indirectly influence prescribing. While we did not have suitable data linkages to identify practices/CCGs near such tertiary referral centres, this could be explored if a reader has specific examples in mind on the OpenPrescribing site

(https://openprescribing.net/analyse/#org=CCG&numIds=0904010AK,0904010IO,0904010HO,0904010LO,0904010JO,0904010ZO,0904010FO,0904010AE,090401050,0904010V0,090401060,0904010E0,0904010T0,0904010AF,090401030,0904010Q0,0904010A0,0904010U0,0904010AD,0904010AC,0904010AB,090401070,0904010AI,0904010AJ,0904010AH,0904010AG,0904010AA,090401080,0904010AU&denom=total_list_size&selectedTab=map).

Reviewer: 2

Please leave your comments for the authors below

I thoroughly enjoyed reading this well written manuscript and commend the authors for undertaking this interesting and important work.

I have 2 minor comments that the authors may wish to consider:

1) I think it would be useful within the Introduction to emphasise this differences in prescribing practices between England and the devolved nations.

- We agree, the devolved nations broadly follow national prescribing guidelines. We have added a section in the introduction to describe this.

2) Please could the authors provide complete details for reference 1.

- These details have now been added.

VERSION 2 – REVIEW

REVIEWER	Federico Biagi University of Pavia, Italy
REVIEW RETURNED	07-Feb-2018
GENERAL COMMENTS	I am now happy with this paper